# The Contribution of Registered Dietitians in the Management of Hyperemesis Gravidarum in the United Kingdom

**DOI:** 10.3390/nu13061964

**Published:** 2021-06-08

**Authors:** Kate Maslin, Hazel A. Billson, Caitlin R. Dean, Julie Abayomi

**Affiliations:** 1School of Nursing and Midwifery, Faculty of Health, University of Plymouth, Plymouth PL4 8AA, UK; 2Department of Nutrition & Dietetics, Liverpool Women’s NHS Foundation Trust, Liverpool L8 7SS, UK; hazel.billson@lwh.nhs.uk; 3Obstetrics and Gynecology, Medical Centre, University of Amsterdam, 1105 AZ Amsterdam, The Netherlands; caitlinrdean@gmail.com; 4Applied Health and Social Care, Edge Hill University, Ormskirk L39 4QP, UK; abayomij@edgehill.ac.uk

**Keywords:** pregnancy sickness, pregnancy nutrition, gestational malnutrition, hyperemesis gravidarum, maternal nutrition

## Abstract

Hyperemesis Gravidarum (HG) is a condition at the extreme end of the pregnancy sickness spectrum, which can cause poor oral intake, malnutrition, dehydration and weight loss. The aim of this study is to explore the role of Registered Dietitians (RD) in the management of HG in the United Kingdom (UK). A survey was designed and distributed electronically to members of the British Dietetic Association. There were 45 respondents, 76% (n = 34) worked in secondary care hospitals, 11% (n = 5) were in maternal health specialist roles. The most commonly used referral criteria was the Malnutrition Universal Screening Tool (40%, n = 18), followed by second admission (36%, n = 16). However 36% (n = 16) reported no specific referral criteria. About 87% (n = 37) of respondents did not have specific clinical guidelines to follow. Oral nutrition supplements were used by 73% (n = 33) either ‘sometimes’ or ‘most of the time’. Enteral and parenteral nutrition were less commonly used. There was an inconsistent use of referral criteria to dietetic services and a lack of specific clinical guidelines and patient resources. Further training for all clinicians and earlier recognition of malnutrition, alongside investment in the role of dietitians were recommended to improve the nutritional care of those with HG.

## 1. Introduction

Nausea and vomiting are reported to be experienced by around 70% of pregnant women [1]. Hyperemesis Gravidarum (HG) is a more severe condition at the extreme end of the pregnancy sickness spectrum. It is estimated to affect 1.5% of pregnant women in the United Kingdom (UK) [2], with rates of 0.8–3.3% reported in other populations [3,4,5,6]. There is no agreed definition of HG, which makes diagnosis, management and research challenging; but HG is typically reported as involving persistent and intractable nausea and vomiting with the triad of more than 5% pre-pregnancy weight loss, dehydration and electrolyte imbalance [7]. The historical belief that HG is self-limiting without any short or long-term consequences has been largely disproven [8]. Symptoms of HG can be persistent, resulting in prolonged poor oral intake, malnutrition, dehydration and weight loss [9,10,11]. Consequently, HG can increase the risk of negative outcomes for the offspring, including preterm birth or being small for gestational age, with growing evidence of the risk for long-term effects also [12,13]. In severe cases, HG can result in Wernicke’s encephalopathy or even maternal death [14]. The exact aetiology of HG is poorly understood, although a genetic origin, associated with cachexia, has been identified [15].

In the absence of a definitive cause, the management of HG focuses on symptom relief and the prevention of serious morbidity [16]. First line advice recommends that women who are vomiting, but are not dehydrated, can be managed in the community with oral antiemetic medication, support, reassurance, oral hydration and dietary advice [7,9]. In practice, hospital treatment with intravenous (IV) fluids, IV antiemetic medication and vitamin B1 supplements is often required. As there is little high quality and consistent evidence supporting any one intervention, effective treatment often requires a combination of medical interventions, dietary and lifestyle changes, supportive care and patient education [9]. Although poor nutritional intake is often a key feature and consequence of HG, there is a lack of research about dietary intake and/or nutritional interventions [17,18,19]. A recent consultation [20], combining views from patients, clinicians and researchers prioritised the ten most important unanswered questions about HG, three of which directly relate to nutritional effects and management.

Those affected by HG can experience micronutrient deficiencies, severe food aversions and poor nutritional intake, in addition to gastroesophageal reflux and gastroparesis [11]. A dietetic consultation may be helpful in assessing and monitoring nutritional status, expanding food choices, and prescribing oral nutritional support where indicated [9]. In some severe cases, more intensive nutritional management using enteral nutrition (EN) may be appropriate. It is recommended that a dietitian should be consulted when tube feeding is being considered [7], although the evidence base is mixed and international practices vary [18,21]. Due to poor intake, unintentional weight loss and electrolyte imbalance, HG may increase the risk of refeeding syndrome. Dietetic advice can be very helpful in advising on its prevention and treatment [9]. Apart from this, the nature and extent of dietetic involvement in the management of HG is not known. The aim of this study is to explore the role of dietetic practice in the management of pregnancies affected by HG in the UK, with a focus on referral criteria, clinical management and strategies to improve nutritional outcomes.

## 2. Materials and Methods

### 2.1. Participants and Recruitment

Participants were Registered Dietitians practising in the UK who had ever had a patient referred to them with suspected or confirmed HG. A link to the information sheet and electronic questionnaire (see Appendix A) was distributed to members of the British Dietetic Association (BDA). Data collection took place for six weeks during November–December 2020 using the ‘Online Surveys’ platform. As this was an exploratory study, a sample size calculation was not required.

### 2.2. Survey Design and Piloting

A short questionnaire was developed based on the study objectives following a comprehensive literature review. It was piloted amongst a small group of dietitians and a patient advocate and then adjusted accordingly. Changes included asking more details about the use of enteral and parenteral nutrition.

### 2.3. Data Analysis and Storage

Data were exported from Online Surveys to Statistical Package for the Social Sciences version 24.0 [22]. Normality of data was assessed, showing data were non-normally distributed. Quantitative data were analysed using descriptive statistics and frequencies. Correlations were undertaken using a one-sided Spearman’s rho. Statistical significance was assumed at *p* < 0.05. Responses to open ended question were assessed and categorised into themes using Braun and Clarke’s step by step guide [23].

## 3. Results

### 3.1. Participant Characteristics, Clinical Setting and Role

Total of 45 participants completed the survey, of which the majority worked in a secondary care hospitals in England. Only 5 (11%) respondents had a principal role as a maternal health dietitian. Further details are shown in Table 1.

When asked to rate their competence in the dietetic management of HG on a scale of 1 (not at all competent) to 10 (very competent), the majority of respondents (60%, n = 27) rated their competence as ≥6. Respondents received a median of 3 (IQR 5.3) referrals per year, however these data were very skewed, with a wide range of 1–100. Six (13%) respondents received ≥25 referrals per year. There was a positive correlation between perceived competence and the number of referrals received per year (Spearmann’s rho 0.307, *p* < 0.05).

### 3.2. Referral Pathway and Criteria

Midwives were the most likely source of referrals (64%, n = 29), followed by obstetricians (42%, n = 19) and nurses (40%, n = 18). The criteria used to refer a patient with HG to a dietitian are shown in Figure 1. Participants were asked to tick all criteria that applied. There were 70 responses, of which the most frequently used referral criteria was the Malnutrition Universal Screening Tool (MUST) (26%, n = 18), see Figure 1 for details. The ‘other’ category (13% (n = 9) responses) included ‘patients not able to tolerate oral food’, ‘non-specific clinical judgement by ward staff’ and ‘placement of nasogastric tube’. For those who responded that ‘percentage weight loss’ was used; 5 or 10% was the stated threshold.

Participants were able to select as many answers as applied in their work setting, therefore the total number of responses exceeds the number of participants.

### 3.3. Clinical Management

An overwhelming majority of respondents (87%, n = 39) reported that no specific clinical guidelines were used in the dietetic management of HG. Only 6 (13%) respondents used either a local or national guideline, most commonly the Royal College of Obstetricians & Gynaecology (RCOG) [7] guidance. Refeeding syndrome guidelines were followed ‘most of the time’ (66.7%, n = 30) or ‘some of the time’ (12%, n = 6). A fifth of respondents (20%, n = 9) selected that refeeding syndrome was ‘not usually clinically relevant’ in their caseload of those with HG.

### 3.4. Use of Oral Supplements, Enteral and Parenteral Nutrition Support

Figure 2 shows how frequently different types of nutrition support are typically used. Oral nutrition supplements (ONS) were most commonly used, with 31% (n = 14) of respondents stating they were used ‘most of the time’ and 42% (n = 19) selecting ‘sometimes’. In contrast, 38% (n = 17) and 62% (n = 28) reported that enteral and parenteral nutrition respectively were never used. Where EF was used, the nasogastric route (63%, n = 17) was more common than nasojejunal feeding (37%, n = 10).

The most commonly selected contraindications to parenteral nutrition (PN) from a predefined list of responses were: ‘unlikely to need PN for a sufficient time for risks to outweigh the benefits’ (50%, n = 20) and ‘lack of training in maternity unit’ (45%, n = 18). ‘Risk of infection’ (20%, n = 8) and ‘lack of funding/service’ (15%, n = 6) were also relevant factors.

When respondents were asked to select from a list of what they would find most helpful in supporting those with HG, most (96%, n = 43) reported they would find specific HG resources useful, with many also selecting ‘more training for dietitians’ (78%, n = 35), including webinars (69%, n = 31) and ‘more training for other health care professionals’ (HCPs) (62%, n = 28).

### 3.5. Improvement of Care

When asked what would help optimise the referral rate and dietetic management of pregnancies with HG, 26 individuals gave free text responses. Following categorisation, three key themes emerged, namely:A need to increase awareness and training for healthcare professionals;Improved guidance for dietetic referral and management of HG;Improved capacity/priority for dietitians to cope with the issue.

Illustrative quotes for these themes (and sub-themes) are shown in Table 2.

## 4. Discussion

This study aimed to explore dietetic involvement in the management of HG; to our knowledge, the first to research this topic. The results from 45 respondents demonstrate a lack of specific nutritional guidelines and patient resources for this patient group, with inconsistency in nutritional screening and referral practices. The numbers of patients seen per year was very low, although this varied considerably, with only a small minority of respondents having a specialist maternal health role. Although EN and PN were reported to be used infrequently, ONS were used more regularly. The key themes for improvement of dietetic referral and management show that participants were keenly aware of the nutritional risk to women with HG and the need for nutritional care to improve.

The most commonly used referral criteria was via MUST (40%, n = 18), a tool based on a composite score derived from three independent criteria: current body mass index (BMI), (unintentional) weight loss and acute disease effect score. The acute disease effect component is defined as ‘acutely ill and where there has been or is likely to be no nutritional intake for >5 days’. Meeting this criteria alone, equates to a ‘high risk for malnutrition’, with the algorithm recommending referral to a dietitian [24]. Therefore even without incurring any score for the other two criteria, a person experiencing severe HG symptoms with minimal oral intake, would be classified as high risk. MUST is valid, reliable and authors recommend it can be applied to pregnancy with cautious interpretation [24]. Authors recommend that recalled weight could be used to estimate pre-pregnancy BMI category and that weight gain of <1 kg (<0.5 kg in those with obesity) during the 2nd and 3rd trimester generally requires further evaluation [24]. From this survey, it is unclear if MUST was adapted for pregnancy. Other authors suggest a pragmatic threshold of 8–10% pre-pregnancy weight loss warrants further nutritional intervention [9], although in our study weight loss was not a common referral criteria (7%, n = 3).

Regarding other referral criteria, the second most common was ‘being referred after one admission with vomiting’ (36%, n = 16). This may mean that those who are admitted once are never referred to a dietitian, and no objective nutritional screening is undertaken. An analysis of hospital admission data in England between 1998 and 2012 [4], found the readmission rate for HG was 28%, with 11% having ≥3 admissions. This implies that 72% would not meet the threshold for dietetic referral, even with significant weight loss and/or poor oral intake. However hospital admissions and symptom severity are not always correlated and admissions may be partly related to demographic factors, with some preferring to stay at home [4]. Qualitative research has also found that many of those with HG do not access treatment, which can be attributed to a number of factors, including stigma [25]. Conversely, an analysis of >400,000 pregnancies in England concluded that the low-level prescription of anti-emetics in primary care, may mean that women who could be well managed in the community, present for hospital admission [4]. Therefore hospital readmission is not likely to be a valid indicator of nutritional risk.

Some 36% (n = 16) of responses indicated that ‘no specific referral criteria was used’ thus underlining that better and more consistent nutritional screening is needed. Nutritional screening is a rapid, simple and general procedure used by nursing, medical or other healthcare staff, often at first contact with the patient, to detect those with significant risk of nutritional problems, so that clear guidelines for action can be implemented [24]. Improved training of medical and nursing staff in nutritional screening was recommended by our respondents. However, the physiological changes that occur during pregnancy mean standard malnutrition assessment tools and biochemical reference ranges used in the adult non-pregnant population are not always appropriate, with specific definitions and criteria for malnutrition during pregnancy or ‘gestational malnutrition’ lacking from current international guidelines [26,27]. As such, a recent systematic review concluded that more research is needed to examine the validity and reliability of screening/assessment tools in identifying malnutrition in pregnancy [28].

A more detailed integrated dietetic pathway with thresholds of how to escalate nutritional care is warranted. This was a major theme when respondents were asked what would improve dietetic management. Anecdotally, first line dietary advice has included avoiding fatty/odorous foods/eating on an empty stomach, eating dry crackers and/or eating a high-protein snack before bed [29,30]; which may provide some symptomatic relief in those with mild/moderate symptoms [31,32]. However, there has been no evidence-based research on the effectiveness of these approaches or any other dietary or lifestyle interventions in HG [19]. For the severe symptoms of HG, lifestyle and dietary changes alone are insufficient [8]. This is particularly relevant to our respondents, the vast majority of whom work in hospitals. Of note there has been little research conducted on the use of ONS in HG, which may be a useful nutritional treatment option, given that 73% (n = 33) of our respondents used ONS either ‘sometimes’ or ‘most of the time’.

The results showed that nasogastric feeding was more common than the nasojejunal route. NJT placement is less straightforward than NGT as it requires placement of the distal end of the feeding tube beyond the gastric pyloric sphincter, however it may be less likely to be dislodged by vomiting. Overall the evidence base for the use of tube feeding in HG is inconsistent. A retrospective case series of 558 hospitalised women in Norway, found that compared with other fluid/nutrition regimens, EN was a feasible treatment option, associated with adequate maternal weight gain and favourable pregnancy outcomes [21]. In contrast, a randomised controlled trial based in the Netherlands, that compared EN for 7 days to standard care, did not find any improvement in birth weight or secondary outcomes [18]. Of note, there was a poor protocol completion in the group allocated to EN, with ~7% refusing tube placement, and an unanticipated 34% discontinuing EN because of adverse effects. Conversely, a small qualitative study of 13 women [25], found that those without EN (n = 5), expressed a desire for EN, in order to prevent severe weight loss, dehydration and weakness, to provide sufficient nutrition for the baby and to prevent further admissions. In the UK, EN is viewed as an effective, but extreme method, often used as a last resort. However some of the responses to the survey suggested that NG/NJT should be sited proactively rather than waiting for weight loss to worsen. This is a common strategy in other clinical scenarios in those at risk of malnutrition and where there is insufficient nutrition taken via the oral route [27]. The finding that only 11% (n = 5) used total PN ‘sometimes’, is reflective of guidance that it should only be used in refractory cases when ‘all other medical therapies have failed’.

This study has a number of limitations. Due to the small sample size, inferential statistics were not possible. Attempts were made to maximise recruitment, by cascading the questionnaire, however equally, the small sample size may be reflective of the low numbers of referrals made to dietitians for pregnancies affected by HG, rather than a low response rate. Previous UK research reported that dietetic resources for specialist maternity services were inadequate despite dietitians recognising the importance of nutritional counselling during pregnancy [33]. Responses were based on self-report, rather than clinical records. The survey used lists of predefined responses, however the responses were developed and piloted with dietitians. As the study was UK-based, it may not be externally generalisable. Strengths of the study are the variety of participants working in different roles, settings and the four countries of the UK.

## 5. Conclusions

HG is an extremely debilitating condition that often requires hospital admission and can have severe nutritional consequences for both maternal and fetal health. Our results indicate there is currently an inconsistent approach in referring those with HG for dietetic input, combined with a lack of nutritional-specific guidelines and pathways. Improved training and education of all HCPs and the development of evidence-based nutritional guidelines, with enhanced funding for dietitians were suggested ways to improve care to this population group.

## Figures and Tables

**Figure 1 nutrients-13-01964-f001:**
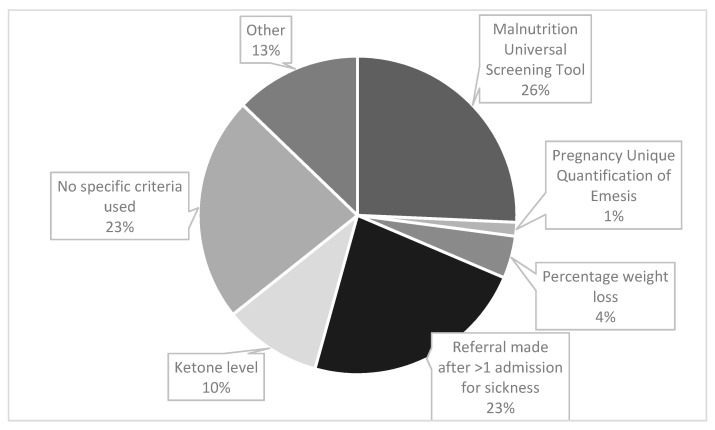
Clinical criteria used to refer a patient with HG for dietetic input.

**Figure 2 nutrients-13-01964-f002:**
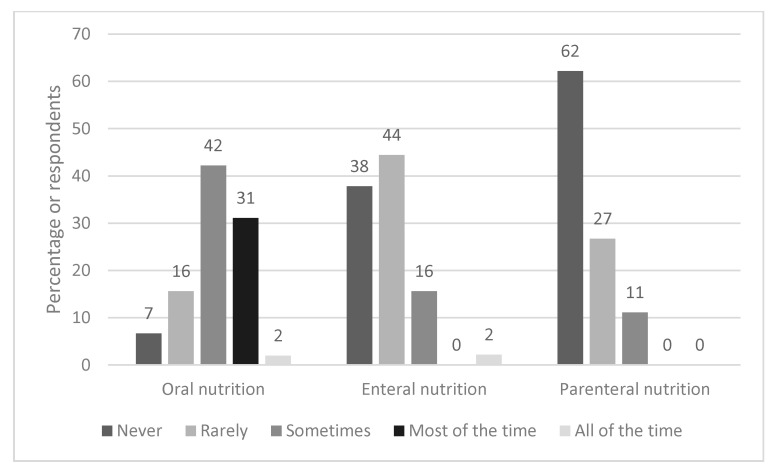
Frequency and type of nutrition support used.

**Table 1 nutrients-13-01964-t001:** Respondent characteristics.

	Response Options	% (n)
Clinical setting	Secondary care NHS hospital Tertiary care NHS hospital Specialist NHS maternity hospital Private hospital Other	76 (34) 11 (5) 4 (2) 0 (0) 9 (4)
Country	England Scotland Wales Northern Ireland	64 (29) 11 (5) 9 (4) 16 (7)
Work experience as a dietitian	<2 years 2–5 years 6–10 years >10 years	16 (7) 24 (11) 9 (4) 51 (23)
Main role/specialism	General/newly qualified Gastroenterology Nutrition Support Team Maternal health dietitian Other ^1^	20 (9) 16 (7) 20 (9) 11 (5) 36 (16)
Service level agreement to provide dietetic cover to maternity/obstetrics patients	Yes No Don’t know	40 (18) 42 (19) 18 (8)
Setting where HG patients usually seen	Inpatient Outpatient Day case setting Other	78 (35) 40 (18) 13 (6) 4 (2)
Access to day unit rehydration service for HG patients in place of work	Yes No Not sure	18 (8) 44 (20) 38 (17)

^1^ Other main roles included diabetes, dietetic manager, critical care, inherited metabolic disease. NHS: National Health Service. HG: Hyperemesis Gravidarum.

**Table 2 nutrients-13-01964-t002:** Themes from responses to question: ‘What would help optimise the referral rate and dietetic management of pregnancies with HG?’

Theme	Sub-Theme	Example Quotes
1. A need to increase awareness & training for HCPs	Need to increase awareness amongst HCPs	‘I saw a patient this week and still this idea persists, where the referral stated “it might be psychological”—this woman had lost >6 kg in 7–8 weeks’. ‘Acknowledgement that this is very real and debilitating’. ‘Raising awareness amongst ward staff’. ‘More awareness of how serious this condition is’. ‘Insight into refeeding risk’. ‘Prompt siting of NJT/PN consideration if vomiting uncontrolled or weight loss is severe’.
	Improved training of HCPs	‘Training of multidisciplinary teams (MDT) seeing patients’. ‘Teaching of doctors and student dietitians’. ‘Training for staff that these patients are nutritionally at risk’. ‘Understanding the value of nutritional supplements’. ‘Building good working relationships with midwives on antenatal wards’.
2. Improved guidance for dietetic referral and management of HG	Improved guidance for referral for dietetic input	‘MUST … not always done but is our acceptance criteria’. ‘Biochemistry results’. ‘Screened for % weight loss and length of time with little or no nutrition’. ‘Tools available to screen patients and agreed referral criteria’. ‘Referral to dietitians on admission…instead of waiting for weight loss’.
	Improved guidance for dietetic management	‘Specific evidence based guidelines that we could share with maternity services’. ‘Treatment is limited to repeated admissions for rehydration and anti-emetics’. ‘More specialised service to refer women to and updated literature’. ‘Increased training of dietetic management of HG’. ‘Information leaflets on HG and some guidelines to follow e.g., vitamins to consider…or a list of services…to refer the patient on’.
3. Improved capacity/priority for dietitians to cope with the issue		‘Funding for dietetics, research into impact and benefits of dietetics’. ‘We don’t have the capacity to advertise dietetics to this group of service users’. ‘Increased profile of dietetics among other HCPs’. ‘If people thought dietitians could do more’. ‘Specific funding for maternity care’.

HCP: Health Care Professionals → NJT: Nasojejunal Tube → PN: Parenteral Nutrition → MUST: Malnutrition Universal Screening Tool HG: Hyperemesis Gravidarum.

## Data Availability

The data presented in this study are available on request from the corresponding author.

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
