# Peer review of "The Contribution of Registered Dietitians in the Management of Hyperemesis Gravidarum in the United Kingdom"

_nutrients, 2021, doi:10.3390/nu13061964_

Round 1

Reviewer 1 Report

The authors have conducted a survey where nutritionists were asked questions regarding their role in treating patients with hyperemesis gravidarum. Nutrition is an important, and often neglected, part of hyperemesis treatment. The fact that only 45 nutritionists answered (I miss a figure regarding potential responders; numbers of Registered Dietitians practising in the UK or perhaps numbers of members in the British Dietetic Association (BDA)) illustrates this fact. ? I suggest the authors use this probably low ratio responders vs potential participants to discuss whether this also represents a general lack of focus of hyperemesis as causing nutritional deficiencies.

Also regarding nutritionists intervention; I miss a point regarding evidence of possible interventions to reduce nausea and vomiting in pregnancy. To develop guidelines this knowledge should be presented. If this information is unavailable, further studies are needed to identify such interventions/types of food to recommend.

Otherwise the presentation is good, linguistic, type of analyses performed and result presentation.

Minor;

With a cohort of 45, the percentages should be presented as whole numbers, not decimals. Do not state data as non-parametric but not normally distributed.

Statistical test were they one- or two-sided?

In tables/figures explain ALL abbreviations (e.g. HG and NHS) in EACH table/figure.

Reviewer 2 Report

The manuscript by Maslin et al with the title “The contribution of Registered Dietitians in the management of Hyperemesis Gravidarum in the United Kingdom” reports results from a survey among dietitians in the UK on nutritional care of patients with hyperemesis gravidarum (HG). The survey contained highly relevant questions for investigating the impact of nutrition on patients with HG. The results showed that there was a lack of specific guidelines to follow, and inconsistent use of referral criteria, so the authors suggested several areas of improvement in order to advance the nutritional care of those with HG, including training of clinicians and better recognition of malnutrition. This subject area should be of interest to the readers of Nutrients since it pinpoints an area of concern with regard to nutrition and human health. My recommendation is therefore to accept the manuscript for publication. Since essentially no revision is needed, only some minor editing to polish the manuscript, publication in present form is a possibility for this manuscript. Please see below for the few concerns of minor importance that I have. • 2.2. Survey design and piloting ”It was piloted amongst a small group of dietitians and a patient advocate and then adjusted accordingly.” It would be interesting to know what adjustments were made. Maybe it would be useful to give a few examples to illustrate the difference between the pilot and final survey. • 2.3. Data analysis and storage - ”.” is missing in ”…showing data was non-parametric”. - I believe it is called Spearman’s rho. • Discussion - “An analysis of hospital admission data in England between 1998-201”, needs to be corrected for the range of years. - One “is” needs to be removed: “…where there is insufficient nutrition is taken via the oral route [32].” Additional comment: Number of digits throughout manuscript and in the table and figure is not consistent. It would be worth it to consider value figures (two is probably most useful in this context).
